# Loss of IP3R-BK_Ca_ Coupling Is Involved in Vascular Remodeling in Spontaneously Hypertensive Rats

**DOI:** 10.3390/ijms241310903

**Published:** 2023-06-30

**Authors:** Sayeman Islam Niloy, Yue Shen, Lirong Guo, Stephen T. O’Rourke, Chengwen Sun

**Affiliations:** 1Department of Pharmaceutical Sciences, North Dakota State University, Fargo, ND 58105, USAyue.shen.1@ndsu.edu (Y.S.); stephen.orourke@ndsu.edu (S.T.O.); 2School of Nursing, Jilin University, Changchun 130021, China; lirong.guo@ndsu.edu

**Keywords:** hypertension, IP3 receptor, BK_Ca_ channel, vascular remodeling, vascular smooth muscle cells

## Abstract

Mechanisms by which BK_Ca_ (large-conductance calcium-sensitive potassium) channels are involved in vascular remodeling in hypertension are not fully understood. Vascular smooth muscle cell (VSMC) proliferation and vascular morphology were compared between hypertensive and normotensive rats. BK_Ca_ channel activity, protein expression, and interaction with IP3R (inositol 1,4,5-trisphosphate receptor) were examined using patch clamp, Western blot analysis, and coimmunoprecipitation. On inside-out patches of VSMCs, the Ca^2+^-sensitivity and voltage-dependence of BK_Ca_ channels were similar between hypertensive and normotensive rats. In whole-cell patch clamp configuration, treatment of cells with the IP3R agonist, Adenophostin A (AdA), significantly increased BK_Ca_ channel currents in VSMCs of both strains of rats, suggesting IP3R-BK_Ca_ coupling; however, the AdA-induced increases in BK_Ca_ currents were attenuated in VSMCs of hypertensive rats, indicating possible IP3R-BK_Ca_ decoupling, causing BK_Ca_ dysfunction. Co-immunoprecipitation and Western blot analysis demonstrated that BK_Ca_ and IP3R proteins were associated together in VSMCs; however, the association of BK_Ca_ and IP3R proteins was dramatically reduced in VSMCs of hypertensive rats. Genetic disruption of IP3R-BK_Ca_ coupling using junctophilin-2 shRNA dramatically augmented Ang II-induced proliferation in VSMCs of normotensive rats. Subcutaneous infusion of NS1619, a BK_Ca_ opener, to reverse BK_Ca_ dysfunction caused by IP3R-BK_Ca_ decoupling significantly attenuated vascular hypertrophy in hypertensive rats. In summary, the data from this study demonstrate that loss of IP3R-BK_Ca_ coupling in VSMCs induces BK_Ca_ channel dysfunction, enhances VSMC proliferation, and thus, may contribute to vascular hypertrophy in hypertension.

## 1. Introduction

Hypertension is a major risk factor for many fatal cardiovascular complications, such as stroke and myocardial infarction, which are currently a leading cause of death in the US and worldwide [1,2]. Although conventional antihypertensive drugs have been successfully used in the clinic to control blood pressure, about 15% of hypertensive patients suffer from drug-resistant hypertension [3]. Patients with resistant hypertension are at high risk of cardiovascular complications and mortality with few treatment options [3]. Therefore, it is urgent to identify novel therapeutic targets in the cardiovascular system for treatment of hypertension.

Hypertension is characterized by enhanced vascular constriction and remodeling [2]. Vascular remodeling, such as hypertrophy and stiffness, is a major pathological alteration associated with chronic hypertension, causing end-organ damage and leading to cardiovascular comorbidities [4]. Vascular smooth muscle cells (VSMCs) play an important role in vascular remodeling, switching their phenotype from the differentiated contractile state to de-differentiated synthetic and proliferative states [5]. Therefore, it is crucial to identify the regulatory mechanisms underlying VSMC phenotypic switching in order to prevent vascular remodeling, end-organ damage, and cardiovascular comorbidities in hypertension.

Large-conductance calcium-activated potassium (BK_Ca_) channels are expressed in the plasma membrane (PM) of VSMCs and carry large-conductance outward potassium currents. The activity of BK_Ca_ is regulated by membrane potential and intracellular Ca^2+^ concentration [6]. Activation of this channel induces membrane potential hyperpolarization and voltage-sensitive L-type Ca^2+^ channel inactivation in VSMCs, leading to vascular smooth muscle relaxation and vasodilation [6]. Conversely, genetic deletion or inhibition of this channel causes membrane depolarization and increases in intracellular Ca^2+^ concentration, which regulates contractile responses, gene expression, cell proliferation, and other cellular functions [7,8,9]. Recent studies have demonstrated that reduced BK_Ca_ channel function in VSMCs is involved in the pathophysiology of hypertension, including increased arterial constriction and elevated cell-proliferation in response to vasoactive mediators, such as norepinephrine (NE) and angiotensin II (Ang II) [7,10]. Therefore, it has been proposed that reduced expression, post-translational modification, and membrane-trafficking of this channel protein may contribute to those vascular alterations in hypertension [11,12,13]. However, the exact intracellular mechanisms involved in BK_Ca_ dysfunction in the pathogenesis of hypertension are not yet fully clear.

In this study, we investigated and compared the protein expression, functional alterations, and intracellular molecular regulation of the BK_Ca_ channel in hypertensive and normotensive animal models. Our goal was to identify the molecular mechanisms underlying BK_Ca_ channel alteration in VSMC phenotype switching and in the development of vascular remodeling during hypertension. By using multiple techniques, we found that the expression of the BK_Ca_ channel, as well as its Ca^2+^-sensitivity and voltage-dependency, were unchanged in VSMCs of hypertensive rats compared to normotensive rats. However, we did observe a reduction in the interaction between the BK_Ca_ channel in the plasma membrane (PM) and the inositol 1,4,5-trisphosphate receptor (IP3R) in the sarcoplasmic reticulum (SR) of VSMCs in hypertensive rats. We further investigated the role of this reduced interaction between IP3R-BK_Ca_ in VSMC proliferation and vascular hypertrophy in hypertension.

## 2. Results

### 2.1. Ca^2+^-Sensitivity of BK_Ca_ Channel Is Comparable in VSMCs from SHR and WKY Rats

Previous studies have demonstrated that reduced BK_Ca_ channel activity contributes to vascular hypercontractility and remodeling in hypertension [7]. Therefore, we examined the possibility that the dysfunction of BK_Ca_ channels is due to altered Ca^2+^-sensitivity in VSMCs of SHR. Single BK_Ca_ channel recording was performed using patch clamp in VSMCs dissociated from mesenteric arteries of SHR and WKY rats, as described in the Methods. BK_Ca_ channel currents were recorded from inside-out patches of VSMCs bathed with different Ca^2+^ concentrations from 0.3 to 3 µM. The results are presented in Figure 1, demonstrating that Ca^2+^ in the bath solution did not alter the unitary amplitude of single-channel currents (Figure 1A,B) at concentrations from 0.3 to 3.0 μM in VSMCs from both SHR and WKY rats. However, the open probability of the BK_Ca_ channel was significantly increased by Ca^2+^ in a concentration-dependent manner in VSMCs from both strains of rats (Figure 1C). More interestingly, the channel activity at each Ca^2+^ concentration was comparable between SHR and WKY rats (Figure 1C). Taken together, the results from this experiment indicate that BK_Ca_ activity is stimulated by intracellular Ca^2+^ and that the Ca^2+^-sensitivity of BK_Ca_ is not significantly altered in VSMCs of SHR as compared with WKY rats.

### 2.2. Voltage-Sensitivity of BK_Ca_ Channels Is Comparable in VSMCs from SHR and WKY Rats

In the next experiment, we examined the possibility that the dysfunction of BK_Ca_ channels is caused by altered voltage-sensitivity in VSMCs of SHR. Single BK_Ca_ channel currents were recorded on inside-out patches of VSMCs at different membrane potentials from −70 mV to +70 mV. The results are presented in Figure 2, demonstrating that the amplitude of the BK_Ca_ channel current was membrane potential-dependent (Figure 2A). The conductance of BK_Ca_ channels was 210 ± 2 pS and 211 ± 3 pS in VSMCs from WKY and SHR, respectively (Figure 2C). The channel open-state probability (NPo) was increased by membrane depolarization (Figure 2A,B). More interestingly, both channel open probability and conductance at each membrane potential were comparable in VSMCs from SHR and WKY rats. In summary, these data demonstrate that the voltage-sensitivity of the BK_Ca_ channel in VSMCs is not significantly altered in SHR as compared with WKY rats.

### 2.3. Functional Coupling between IP3R-BK_Ca_ Is Impaired in VSMCs of SHR

Recently, inositol 1,4,5-trisphosphate receptors (IP3Rs) have been identified in the sarcoplasmic reticulum (SR, an intracellular Ca^2+^ store) mediating Ca^2+^ release from SR after cellular exposure to vasoactive agonists, such as NE and Ang II [14,15,16]. Considering that the activity of BK_Ca_ is regulated by intracellular Ca^2+^, BK_Ca_ channel activity could be maintained by Ca^2+^ released from SR through IP3R. This hypothesis was tested in VSMCs isolated from mesenteric arteries using the whole-cell recording configuration, as described in the Methods. Whole-cell BK_Ca_ currents were recorded from VSMCs before and after treatment with the IP3R agonist, Adenophostin A (AdA, 1 µM). AdA is a hydrophilic compound, not cell-permeable. Therefore, AdA was encapsulated into liposomes as a vehicle for delivery into cells. The drug-delivery liposome preparation has been characterized using fluorescence in our previous study [17]. The results are presented in Figure 3, demonstrating that treatment of VSMCs from WKY rats with AdA significantly increased BK_Ca_ channel density from 62.8 ± 9.6 pA/pF to 142.3 ± 30.0 pA/pF at +60 mV holding potential (*n* = 9 cells, *p* < 0.05; Figure 3A,B). In addition, the liposome vehicle did not significantly alter BK_Ca_ channel currents. This result indicates that the BK_Ca_ channel is activated by an IP3R agonist, likely through Ca^2+^ release, suggesting the functional coupling of IP3R-BK_Ca_ in VSMCs of WKY rats.

We next hypothesized that the dysfunction of BK_Ca_ channels in hypertension is caused by impairment of IP3R-BK_Ca_ coupling in VSMCs. To test this hypothesis, whole-cell BK_Ca_ currents were recorded in VSMCs of SHR under the same treatment conditions as described above. The results are presented in Figure 3C,D, demonstrating that treatment with the IP3R agonist, AdA, also increased BK_Ca_ current density from 50.9 ± 3.2 pA/pF to 65.6 ± 7.1 pA/pF (*n* = 10 cells, *p* > 0.05) at +60 mV holding potential. However, the AdA-induced increases in BK_Ca_ current density were significantly reduced by 80% in VSMCs of SHR as compared with WKY rats. Taken together, these data demonstrate that IP3R-BK_Ca_ functional coupling in VSMCs of SHR is significantly impaired as compared with normotensive WKY rats. This IP3R-BK_Ca_ decoupling may contribute to the reduced BK_Ca_ activity seen in VSMCs of hypertensive rats. In order to confirm this hypothesis, we next examined the molecular association between those two proteins in the following experiments.

### 2.4. Molecular Coupling between IP3R-BK_Ca_ Is Impaired in VSMCs of SHR

To examine BK_Ca_ and IP3R in mesenteric arteries of SHR and WKY rats, conventional Western blots were performed using antibodies against BK_Ca_ and IP3R. The results are presented in Figure 4, demonstrating that the expression of both BK_Ca_ and IP3R is comparable in mesenteric arteries from SHR and WKY rats (Figure 4A–D). Representative blots showing the expression of BK_Ca_ (~110 kDa) and IP3R (~240 kDa) in mesenteric arteries of WKY and SHR are presented in Figure 4A and Figure 4C, respectively. Both BK_Ca_ and IP3R protein expression are comparable in mesenteric arteries between SHR and WKY rats (Figure 4B and Figure 4D, respectively). In summary, data from this experiment indicate that protein expression of both BK_Ca_ and IP3R in mesenteric arteries is not altered in SHR as compared with WKY rats.

In the next experiment, BK_Ca_ antibodies were used to pull down BK_Ca_ protein, some of which could be associated with IP3R, in mesenteric artery samples from SHR and WKY rats. IP3R protein levels in the immunoprecipitated samples were detected using standard Western blots with IP3R1-specific antibodies. The results are presented in Figure 4E,F, indicating that IP3R protein levels in the positive control (the corresponding original samples without co-immunoprecipitation) are comparable between SHR and WKY rats. However, the co-immunoprecipitated sample from WKY rats generated a stronger band as compared with immunoprecipitated samples from SHR (Figure 4E). The negative control lane did not generate any band, as expected, since the negative control sample contained empty beads. Taken together, all results strongly imply that the protein levels of BK_Ca_ and IP3R are not altered in mesenteric arteries of SHR, but the molecular coupling between BK_Ca_ and IP3R is disrupted in mesenteric arteries of SHR, as compared with normotensive WKY rats.

### 2.5. Blockade of BK_Ca_ Channel Enhances Ang II-Induced Proliferation in VSMC of WKY Rats, Mimicking the Enhanced Response to Ang II in SHR

To examine the effect of IP3R-BK_Ca_ decoupling on vasoactive agonist-induced vascular remodeling in VSMCs, the BK_Ca_ channel inhibitor, paxilline, was used to mimic the reduced BK_Ca_ activity due to IP3R-BK_Ca_ decoupling in SHR rats. The cell number and proliferation of VSMCs cultured from mesenteric arteries were examined after treatment with the control, Ang II (0.1 μM), and Ang II plus Paxilline (1 μM). The VSMC number was counted under the microscope after immunostaining with a smooth muscle specific antibody against α-smooth muscle actin. The results are presented in Figure 5, demonstrating that treatment with Ang II significantly increased the VSMC number in both SHR and WKY rats (Figure 5A–G). However, the Ang II-induced increases in cell number were enhanced in SHR by ~48% as compared with WKY rats (Figure 5G). Treatment with paxilline significantly augmented this effect of Ang II in WKY, reaching to a similar level of Ang II effect in VSMCs of SHR. In summary, results from this experiment indicate that the Ang II-induced increase in the VSMC number is enhanced in SHR as compared with WKY rats, and that the BK_Ca_ channel blockade significantly enhanced Ang II-induced increases in the VSMC number of WKY rats, mimicking the augmented Ang II response in SHR.

We next examined VSMC proliferation using the WST-1 cell proliferation assay after the cells were treated with the control, Ang II (0.1 μM), and Ang II plus paxilline (1 μM). The results are presented in Figure 5H, demonstrating that Ang II treatment significantly increased the proliferation of VSMCs of both SHR and WKY rats. The Ang II-induced increases in cell proliferation were enhanced in SHR as compared with WKY rats. Co-treatment with paxilline significantly increased Ang II-induced proliferation in WKY, reaching a level of Ang II-induced response similar to that observed in SHR. Taken together, the results from these studies suggest that the effects of Ang II on VSMC proliferation were enhanced in SHR as compared with WKY rats, and that the enhanced Ang II-induced VSMC proliferation in SHR is caused by impaired IP3-BK_Ca_ coupling.

### 2.6. Disrupting IP3R-BK_Ca_ Coupling Augments Ang II-Induced Proliferation in VSMCs of WKY Rats, Mimicking the Effect of Ang II in SHR

Junctophilin 2 (JPH2) is expressed in VSMCs, where it acts as a major tethering protein between the PM-SR conjunction [18,19]. Therefore, we used JPH2-shRNA to knockdown JPH2 expression and to disrupt IP3R-BK_Ca_ coupling in order to detect the role of IP3R-BK_Ca_ coupling in Ang II-induced proliferation in VSMCs. JPH2 expression was examined in VSMCs cultured from mesenteric arteries of WKY rats after treatment with AAV2-JPH2-shRNA or AAV2-SCR-shRNA (control) using real-time PCR. Treatment with AAV2-SCR-shRNA did not alter JPH2 mRNA levels. However, AAV-JPH2-shRNA significantly reduced JPH2 expression (Figure 6A). The cell number was counted after immunostaining with smooth muscle-specific α-actin antibody in VSMCs treated with AAV2-SCR-shRNA or AAV2-JPH2-shRNA plus Ang II or the control. The results are presented in Figure 6B, indicating that knockdown of JPH2 significantly enhanced Ang II-induced increases in cell number by 43% as compared with the scramble control. Next, VSMC proliferation was measured by the WST-1 approach in VSMCs treated with AAV2-SCR-shRNA or AAV2-JPH2-shRNA plus Ang II or the control. The results demonstrated that knockdown of JPH2 significantly enhanced Ang II-induced cell proliferation by 61% as compared with the scramble control. In summary, the results from these experiments demonstrate that knockdown of JPH2 expression dramatically increases Ang II-induced proliferation in VSMCs, suggesting that disruption of IP3R-BK_Ca_ coupling enhances these actions of Ang II, mimicking VSMCs of SHR.

### 2.7. BK_Ca_ Channel Opener Alleviates Vascular Hypertrophy in SHR

In the next in vivo experiment, the BK_Ca_ channel opener, NS1619, was used to functionally bypass the deficient IP3R-BK_Ca_ coupling in SHR. The mesenteric artery morphology was examined using H&E staining in SHR and WKY rats after subcutaneous infusion of NS1619 for 4 weeks. The results are presented in Figure 7, demonstrating that treatment with NS1619 did not significantly alter the mesenteric arterial morphology in WKY rats. However, chronic infusion of NS1619 dramatically attenuated the vascular hypertrophy in SHR. The results from this study suggest that functional bypass of IP3R-BK_Ca_ decoupling using a BK_Ca_ channel activator ameliorates vascular remodeling in SHR.

## 3. Discussion

The present study provides the first evidence that disruption of IP3R-BK_Ca_ coupling in VSMCs is involved in vascular remodeling in hypertension. This conclusion is supported by the following evidence: (1) The Ca^2+^-sensitivity and voltage-dependence of BK_Ca_ channels are comparable in VSMCs from SHR and WKY rats. (2) an IP3R agonist increases BK_Ca_ channel currents, suggesting the functional coupling IP3R-BK_Ca_, and the functional coupling of IP3R-BK_Ca_ is impaired in VSMCs of SHR. (3) Co-immunoprecipitation results indicate that IP3R-BK_Ca_ proteins are associated together, suggesting molecular coupling of IP3R-BK_Ca_. The molecular coupling of IP3R-BK_Ca_ is reduced in mesenteric arteries of SHR. (4) Inhibition of BK_Ca_ channels augments Ang II-induced proliferation in VSMCs of WKY rats, mimicking the enhanced effect of Ang II in SHR. (5) Disrupting IP3-BK_Ca_ coupling using JPH2-shRNA also increases Ang II-induced VSMC proliferation in WKY rats. (6) Treatment with a BK_Ca_ channel opener, to functionally bypass IP3R-BK_Ca_ deficiency, attenuated the vascular hypertrophy in SHR. All together, these results indicate that loss of IP3R-BK_Ca_ coupling may contribute to vascular remodeling in hypertension. However, several concerns need to be addressed.

The first concern raised from this study is how do BK_Ca_ channels regulate the proliferation of VSMCs, leading to vascular remodeling in hypertension? The mechanism involved in BK_Ca_-mediated VSMC proliferation is still not clear. One possibility could be mediated by the control of intracellular Ca^2+^ concentration. It is well-known that BK_Ca_ channels control intracellular Ca^2+^ by a negative feedback signaling pathway to prevent excessive elevation of intracellular Ca^2+^ after vasoactive agonist stimulation [8,20]. BK_Ca_ dysfunction, such as loss of IP3R-BK_Ca_ coupling, should increase the Ca^2+^ response to agonist stimuli, leading to intracellular Ca^2+^ elevation. Intracellular Ca^2+^ not only triggers vasocontraction, but also regulates proliferation-related gene expression in VSMCs [8], thereby controlling the VSMC phenotype. It has been reported that intracellular Ca^2+^ elevation induces NFAT (nuclear factor of activated T cells) dephosphorylation by activation of calcineurin, a Ca^2+^-dependent phosphatase, in VSMCs [20]. Dephosphorylated NFAT translocates into the nucleus, binding to several transcription coactivators, which in turn lead to VSMC proliferation and vascular remodeling [8,20]. This notion is supported by the current observation that BK_Ca_ channel inhibition or disruption of IP3R-BK_Ca_ coupling significantly increases Ang II-induced VSMC proliferation. This observation is also consistent with other observations showing that the disruption and blockade of BK_Ca_ function induces VSMC proliferation and vascular hypertrophy [7,10]. The results from this study add further supporting evidence by showing that treatment of SHR with a BK_Ca_ opener significantly attenuated the development of vascular remodeling.

The results from the current study show that BK_Ca_ and IP3R couple together to maintain BK_Ca_ channel activity. The next question raised from this study could be how is BK_Ca_ activity regulated by IP3R in VSMCs? IP3R is expressed in the SR, functioning as a Ca^2+^ releasing channel from this intracellular Ca^2+^ store [14,15,16]. The IP3R is a receptor for inositol 1,4,5-trisphosphate (IP3), a plasma membrane lipid produced by phospholipase C (PLC) in response to the stimulation by vasoactive agonists, such as Ang II [21]. The co-immunoprecipitation data in the current study demonstrate that BK_Ca_ and IP3R proteins are associated closely together. Considering BK_Ca_ channel activity is regulated by intracellular Ca^2+^, it is possible that the IP3R-derived Ca^2+^ release from the SR may stimulate BK_Ca_ activity. This hypothesis is supported by our whole-cell patch clamp studies showing that an IP3R agonist increases the BK_Ca_ current. In addition, results from other research groups also show the direct association of IP3R and BK_Ca_ channels in the PM-SR conjunction of VSMCs [14]. However, the detailed molecular interactions in this signaling pathway in VSMCs in response to pathologic stimuli still need further investigation.

Interestingly, the coupling of IP3R-BK_Ca_ is impaired in VSMCs of SHR as compared with WKY rats. Considering that BK_Ca_ channel activation induced by IP3-induced Ca^2+^ release provides an important negative feedback mechanism to protect against overresponse to vasoactive agonists [22,23], loss of coupling between IP3R-BK_Ca_ would induce BK_Ca_ dysfunction, thereby increasing the sensitivity of the blood vessels to agonist stimuli and leading to vascular hypertrophy and hyperplasia. This speculation is supported by data from our current study showing that blockade of BK_Ca_ channels or disruption of IP3R-BK_Ca_ coupling enhances Ang II-induced VSMC proliferation in WKY rats, mimicking the enhanced response to Ang II in SHR. Conversely, treatment with a BK_Ca_ channel opener, bypassing IP3R-BK_Ca_ decoupling, attenuates development of vascular hypertrophy in SHR. In this study, SHR, a genetic hypertension animal model, was used. Of course, all observations from SHR should be confirmed in other hypertensive animal models, as well as in vascular tissues of hypertensive humans in future studies.

While interest in studying the importance of SR-PM conjunction sites has grown tremendously recently, little is known about their role in the development of hypertension. The mechanism involved in IP3R-BK_Ca_ coupling during hypertension is still unknown. One possible mechanism of IP3R-BK_Ca_ decoupling in VSMCs of SHR could be mediated by a deficit of JPH2, the tether protein between SR-PM. This hypothesis is supported by our data showing that knockdown of JPH2 expression enhances Ang II-induced VSMC proliferation. This notion is also supported by previous investigations showing that JPH2-knockout mice develop severe cardiac hypertrophy [24] and that JPH2 could be cleaved by calpain, an intracellular calcium-sensitive protease [25]. Therefore, we can speculate that elevated calcium within VSMCs of SHR stimulates calpain, which cleaves JPH2, leading to IP3R-BK_Ca_ decoupling. However, this hypothesis requires further investigation and will be the focus of our future studies.

In conclusion, results from this study demonstrate that Ca^2+^-sensitivity, voltage-dependence, and protein expression of the BK_Ca_ channel in VSMCs are comparable between SHR and WKY rats. Impaired IP3R-BK_Ca_ coupling is observed in VSMCs of SHR as compared with WKY rats. This IP3R-BK_Ca_ decoupling in VSMCs of SHR may contribute to the vascular hypertrophy and remodeling observed in hypertension.

## 4. Materials and Methods

### 4.1. Animals and Materials

Experiments were performed on 4–6-month-old spontaneously hypertensive (SHR) rats and normotensive Wistar Kyoto (WKY) rats of either sex. The rats were purchased from Charles River (Wilmington, MA, USA) and housed under controlled conditions with a 12:12 h light–dark cycle. Food and water were available to the animals ad libitum. All animal protocols were approved by the Institutional Animal Care and Use Committee (IACUC) of North Dakota State University.

Paxilline, Angiotensin II (Ang II), and NS1619 were purchased from Cayman (Ann Arbor, ML, USA). Anti-SM actin antibody, Dulbecco’s modified Eagle medium (DMEM), and fetal bovine serum (FBS) were obtained from Thermo Fisher Scientific (Waltham, MA, USA). Collagenase was purchased from Worthington Biochemical (Lakewood, NJ, USA). Anti IP3 receptor (IP3R)1 antibody and protein A/G plus agarose were obtained from Santa Cruz (Dallas, TX, USA). *KCNMA1* antibody was purchased from Alomone Lab (Jerusalem, Israel). 4-[3-(4-iodophenyl)-2-(4-nitrophenyl)-2H-5-tetrazolio]-1, 3-benzene disulfonate (WST-1) was obtained from Abcam. 4′,6-diamidino-2-phenylindole (DAPI), Adenophostin A (AdA), Norepinephrine (NE), ATP, GTP, HEPES, and others were purchased from Sigma-Aldrich (St. Louis, MO, USA).

### 4.2. VSMC Isolation and Culture

The VSMCs were dissociated from small mesenteric arteries using enzymatic methods, as described in our previous publication [26]. Briefly, small mesenteric arteries were dissected under a microscope at third- and fourth-order branches, then incubated for 10 min in 2 mL of low-Ca^2+^ Tyrode’s solution containing (in mM) 145 NaCl, 4 KCl, 0.05 CaCl_2_, 1 MgCl_2_, 10 HEPES, and 10 dextrose, plus 1 mg/mL albumin, followed by 20 min at 37 °C in the same Tyrode’s solution containing 1.5 mg/mL papain and 1 mg/mL DTT. Finally, the arterial segments were incubated for 30 min at 37 °C in the Tyrode’s solution containing 2 mg/mL collagenase, 0.5 mg/mL elastase, and 1 mg/mL soybean trypsin inhibitor. Tissues were then triturated gently using a fire-polished wide-bore pipette to release single VSMCs. Cells were either stored in low-Ca^2+^ Tyrode’s solution at 4 °C for electrophysiological experiments within 6 h or preparation of VSMC cultures.

### 4.3. VSMC Culture and Transfection

Dissociated mesenteric arterial VSMCs were cultured in 25 cm^2^ culture flasks, which contained DMEM supplemented with 10% FBS, penicillin (100 U/mL), and streptomycin (100 μg/mL). Cells were passaged as they became confluent, and cells at the 3rd–5th passages were used for experiments. The VSMC marker, α-smooth muscle actin (α-SMA), was used to identify the VSMCs by using immunocytochemistry, as described in our previous publication [27]. The cultured VSMCs were used to study cell morphology and proliferation. To examine the role of IP3R-BK_Ca_ coupling in Ang II-induced alterations in cell morphology and proliferation, AAV2-mediated overexpression of Junctophilin-2 shRNA (AAV2-JPH2-shRNA) was used to knockdown Junctophilin-2 (JPH2) expression. JPH2 is the major tethering protein between the plasma membrane (PM) and sarcoplasmic reticulum (SR), facilitating IP3R-BK_Ca_ coupling [19]. The AAV2-JPH2-shRNA and its scrambled control (AAV2-JPH2-SCR) were constructed and prepared as described in our previous publication [28]. The AAV2-JPH2-shRNA or AAV2-JPH-SCR (all in 1 × 10^10^ genome copies per microliter, 2 µL of AAV2 into 5 mL medium) were added to the culture medium. After three days of transduction, the cells were used for morphology and proliferation studies. The successful knockdown of JPH2 was verified using real-time PCR, as described in our previous publication [29]. Briefly, TaqMan probe specific for rat JPH2 was purchased from Applied Biosystems Inc (Waltham, MA, USA). Real-time PCR was performed in an Applied Biosystem PRISM 700 sequence detection system, according to the protocol provided by the manufacturer. Data were normalized to 18S RNA. In each experiment, samples were analyzed in triplicate.

### 4.4. VSMC Proliferation Assay

Primarily cultured VSMCs from small mesenteric arteries of SHR and WKY rats were seeded onto glass coverslips in 35 mm dishes. After reaching 70% confluency, the cells were serum-starved for 24 h before treatment. Cells were stimulated with either the vehicle (PBS), Ang II (0.1 µM), or Ang II plus paxilline (a BK_Ca_ blocker, 1 μM) for 48 h. Cells were washed with PBS, fixed with 4% paraformaldehyde, permeabilized with 0.1% Triton X-100, and blocked with 2% BSA for 1 h at room temperature. Cells were then stained with mouse anti-α-SMA antibody (1:250) in 0.1% BSA at 4 °C overnight, and then labeled with goat anti-mouse secondary antibody conjugated to Alexa Fluor 594 (1:500). The positive cell numbers were counted under a Carl Zeiss LSM 900 microscope (Carl Zeiss, Baden-Wurttemberg, Germany).

VSMC proliferation was detected using the WST-1 reagent. Cells were plated in a 96-well microplate and grown to 70% confluence. After being serum-starved for 24 h, cells were treated with vehicle (PBS), Ang II (0.1 µM), or Ang II plus paxilline (1 μM) for 24 h. The medium was then replaced with 100 µL of WST-1 (1:10 dilution) in fresh medium, followed by incubation for 3 h. Absorbance was measured using a multifunctional microplate reader (Spectra Max M5, Molecular Devices, San Jose, CA, USA) at 440 nm, with reference wavelength set at 630 nm.

### 4.5. Electrophysiological Recordings

BK_Ca_ channel activity in mesenteric VSMCs was recorded at room temperature, either in whole-cell configuration or from inside-out patches, as described in our previous publications [26,30]. An Axopatch 200B patch-clamp amplifier (Axon Instruments, Burlingame, CA, USA) and pCLAMP 10.0 software (Molecular Devices, San Jose, CA, USA) were used to control voltage-clamp and voltage-pulse generation. Voltage-activated currents were filtered at 1 kHz and digitized at 5 kHz. Leakage current was subtracted digitally. Series resistance and total cell capacitance were obtained by adjusting series resistance and whole-cell capacitance using the Axopatch 200B amplifier. For inside-out patches, the recording pipettes (resistance 5–6 MΩ) were filled with a solution containing (in mM) 145 KCl, 1.8 CaCl_2_, MgCl_2_ 1.1, and 5 HEPES; pH 7.2 (KOH). Cells were bathed in a solution containing (in mM) 145 KCl, 1.1 MgCl_2_, 0.37 CaCl_2_, 10 HEPES, 1 EGTA, and 10 dextrose; pH 7.4 (KOH). Free Ca^2+^ levels on the cytoplasmic face of the membrane were set by adding the calculated ratio of CaCl_2_ and EGTA (using Chelator 1.0 software, Schoenmakers, Nijmen, The Netherlands). BK_Ca_ open-state probability (NPo) and unitary amplitudes of single-channel currents were obtained at different membrane potentials between −70 mV to +70 mV (20 mV steps) in the presence of 0.3, 1, 1.5, or 3 μM Ca^2+^.

In whole-cell voltage-clamp experiments, the bath solution contained (in mM) 145 NaCl, 5.4 KCl, 1.8 CaCl_2_, 1 MgCl_2_, 5 HEPES, and 10 dextrose; pH 7.4 (NaOH). The recording pipettes (resistance 3–4 MΩ) were filled with a solution containing (in mM) 145 KCl, 5 NaCl, 0.37 CaCl_2_, 2 MgCl_2_, 10 HEPES, 1 EGTA, and 7.5 dextrose; pH 7.2 (KOH). Cells were held at −60 mV, and 100-millisecond depolarizing step pulses of 20 mV increments from –40 to +80 mV voltages were applied. The BK_Ca_ current was divided by the capacitance and expressed as current density. Analysis was performed offline using Clampfit 10 (Axon Instruments, Burlingame, CA, USA).

### 4.6. Western Blotting and Immunoprecipitation

BK_Ca_ and IP3R protein levels in rat mesenteric arteries were assessed by Western blot analysis, as in our previous publication [27]. Briefly, proteins were separated on 7.5% polyacrylamide gels by SDS-PAGE and electroblotted onto a nitrocellulose membrane. Membranes were blocked in TBS-T (0.08% Tween) containing 5% milk for 1 h, followed by overnight incubation with rabbit polyclonal anti-KCNMA1 or rabbit polyclonal anti-IP3R1 primary antibodies at 4 °C. After washing with TBS-T, membranes were incubated for 1 h with anti-rabbit horseradish peroxidase–conjugated secondary antibodies. To ensure equal loading, the membranes were re-probed for β-actin after stripping. Membranes were developed using enhanced chemiluminescence (Thermo Fisher Scientific, Waltham, MA, USA), and digital images were obtained using an AGFA CP1000 automatic film processor. Relative protein expression values were obtained by dividing the raw values of BK_Ca_ and IP3R by the raw values of β-actin.

For co-immunoprecipitation, mesenteric arteries were lysed in non-denaturizing cell lysis buffer (Abcam, Cambridge, UK) containing a protease inhibitor mixture (Thermo Fisher Scientific, Waltham, MA, USA). The cell lysate (1.5 mg) was incubated with 8 μg rabbit polyclonal anti-KCNMA1 antibody for 2 h, followed by addition of 20 μL protein A/G PLUS–agarose beads (Santa Cruz Biotechnology, Dallas, TX, USA) for 12 h at 4 °C. After the incubation, samples were spun down and washed three times with PBS. Protein contents were then eluted with 2× SDS sample buffer. Total cell lysate was used as the positive control, while empty beads combined with cell lysate without anti-KCNMA1 antibody was used as the negative control. Samples were analyzed using conventional Western blots with mouse monoclonal anti-IP3R1 primary antibody and horseradish peroxidase–conjugated anti-mouse secondary antibody.

### 4.7. Vascular Morphology Detection

SHR and WKY rats were used to examine the effect of a BK_Ca_ channel opener on vascular remodeling. The BK_Ca_ channel opener, NS1619 (20 µg/kg/day), was administered chronically by subcutaneous infusion via osmotic minipumps (ALZET, Model 2004). The minipump implantation was performed as described in our previous publication [27]. After chronic treatment with vehicle control or NS1619 (20 µg/kg/day) for 4 weeks, SHR and WKY rats were euthanized with overdoses of pentobarbital. The mesentery was isolated and cross-sectioned using a Cryostat (Leica Biosystems). The mesenteric artery sections were fixed in 4% formalin/PBS, followed by staining with hematoxylin and eosin (H&E), as described in our previous publication [27]. The vascular morphology was visualized by a microscope (Olympus, Shinjuku City, Japan). Vascular wall thickness was measured with Infinity Capture and Analysis Software (v6.5.6) to evaluate vascular hypertrophy. The external diameter of the arteries measured was 100–150 µm in all arterial cross-sections.

### 4.8. Data Analysis

Results are expressed as means ± SE. Statistical significance was evaluated by one- or two-way ANOVA, as appropriate, followed by either Newman–Keuls or Bonferroni post hoc analysis, where appropriate. Differences were considered significant at *p* < 0.05. Individual probability values are noted in the figure legends.

## Figures and Tables

**Figure 1 ijms-24-10903-f001:**
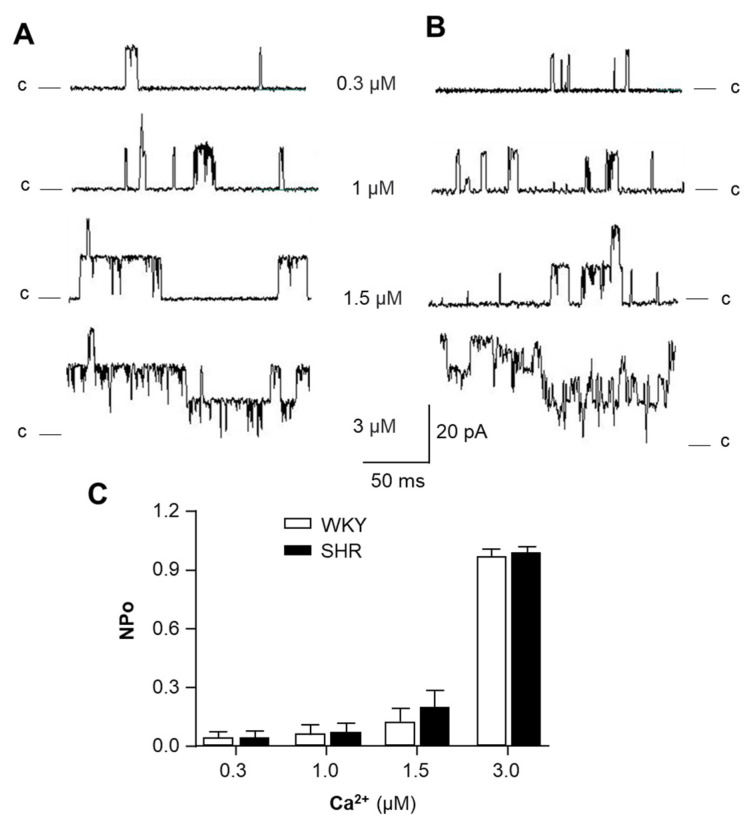
The calcium sensitivity of BK_Ca_ channels in VSMCs of SHR versus WKY rats. BK_Ca_ channel currents were recorded from inside-out patches of VSMCs bathed in symmetrical potassium (145 mM) solution. (**A**,**B**) Representative BK_Ca_ channel current recordings from inside-out patches of VSMCs isolated from mesenteric arteries of WKY rats (**A**) versus SHR (**B**) in the bath solution containing different Ca^2+^ concentrations, as indicated in the figure. (**C**) Bar graphs summarizing the effect of Ca^2+^ on BK_Ca_ channel open probability (NPo) in VSMCs of SHR and WKY rats. Data are mean ± SE, derived from 10 cells from each strain of rats.

**Figure 2 ijms-24-10903-f002:**
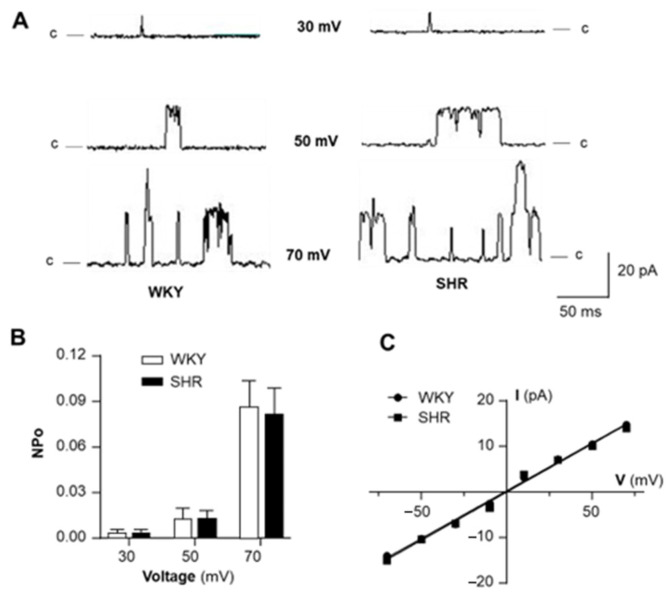
The voltage-dependence of BK_Ca_ channels in VSMCs of SHR versus WKY rats. BK_Ca_ channel currents were recorded from inside-out patches of VSMCs bathed in symmetrical K^+^ (145 mM) solution in the presence of 1 μM Ca^2+^ in the bath solution. (**A**) Representative BK_Ca_ channel current recordings from inside-out patches of VSMCs isolated from mesenteric arteries of SHR (left panel) versus WKY rats (right panel) at holding potentials indicated in the figure. (**B**) Bar graphs summarizing the BK_Ca_ channel open probability (NPo) at holding potentials indicated in the figure. Data are mean ± SE, derived from 6 cells from each strain of rats. (**C**) A summary of current-voltage (I-V) relationships of BK_Ca_ channels in VSMCs from SHR versus WKY rats. Data are mean ± SE, derived from 6 cells from each stain of rats.

**Figure 3 ijms-24-10903-f003:**
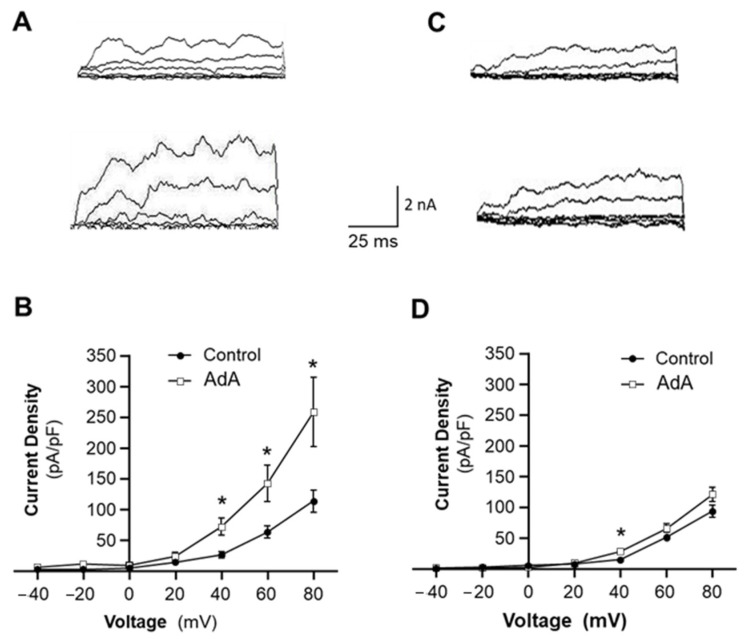
Effects of an IP3R agonist on BK_Ca_ channel currents in VSMCs of SHR and WKY rats. Whole-cell BK_Ca_ currents were recorded at room temperature in response to successive voltage pulses of 100 ms duration, increasing in 20 mV increments from −40 mV to +80 mV in VSMCs before and after treatment with Adenophostin A (AdA, 5 μm), an agonist of IP3R. (**A**,**C**) Representative tracings of BK_Ca_ currents recorded from a single VSMC before (upper panel) and after (lower panel) treatment with AdA in VSMCs isolated from WKY rats (**A**) and SHR (**C**). (**B**,**D**) I-V curves of BK_Ca_ channel activation before and after treatment with AdA in VSMCs isolated from WKY rats (**B**) and SHR (**D**). Data are mean ± SE, derived from 6–10 cells. * *p* < 0.05 as compared with control at corresponding voltage.

**Figure 4 ijms-24-10903-f004:**
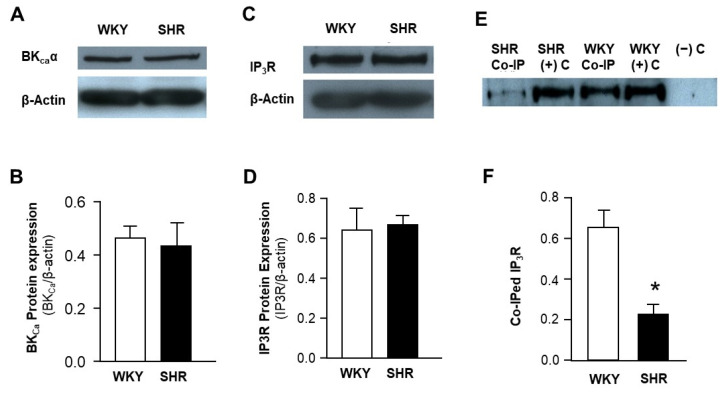
Protein expression of IP3R and BK_Ca_ channels, and their associations in mesenteric arteries of SHR and WKY rats. (**A**,**C**) Representative blots showing protein expression of BK_Ca_α and IP3R1 in mesenteric arterioles dissected from SHR and WKY rats. (**B**,**D**) Bar graphs summarizing the quantification of BK_Ca_α (**B**) and IP3R1 (**D**) protein levels in mesenteric arteries of SHR and WKY rats. Data are mean ± SE, derived from 3 experiments and 6 rats from each strain of rats. (**E**) Representative blots showing IP3R1 band detected in mesenteric arteries after coimmunoprecipitation (Co-IP) with BK_Ca_α antibodies in SHR and WKY rats. Total arteriolar lysate was used as a positive control ((+) C). Empty beads incubated with tissue lysate without BK_Ca_α antibodies as a negative control ((−) C). (**F**) Bar graphs summarizing IP3R1 protein levels after coimmunoprecipitation with BK_Ca_α antibodies in mesenteric arteries of SHR and WKY rats. Data are mean ± SE, derived from 3 experiments and 6 rats from each strain of rats. * *p* < 0.05, significantly different from WKY rats.

**Figure 5 ijms-24-10903-f005:**
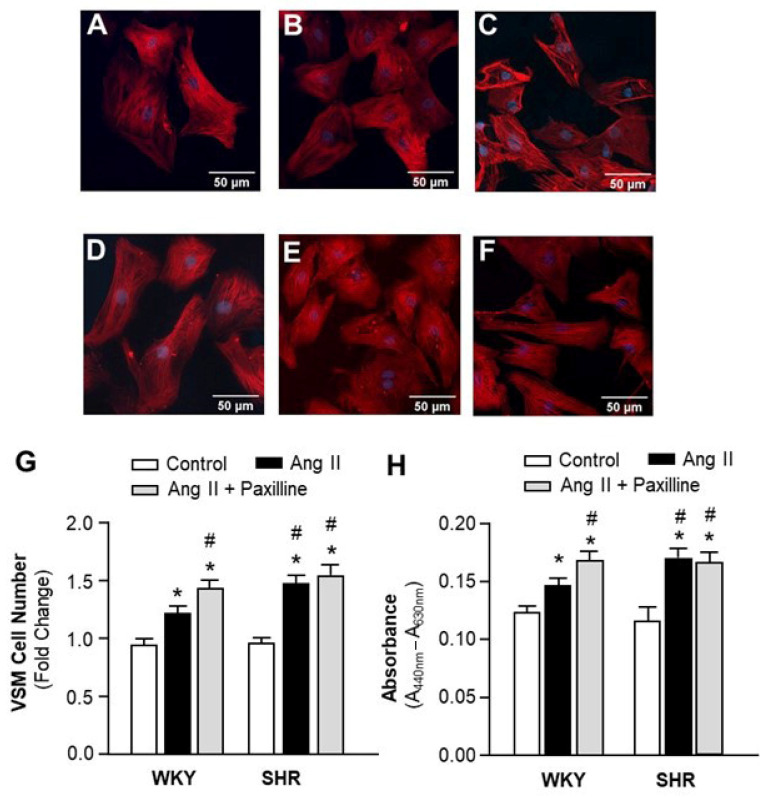
Effect of BK_Ca_ blockade on Ang II-induced proliferation in VSMCs from hypertensive versus normotensive rats. The VSMCs were cultured from mesenteric arteries of SHR and WKY rats. (**A**–**F**) Representative fluorescence micrographs of cells stained with smooth muscle-specific α-actin antibodies after being treated with the control, Ang II (0.1 µM), or Ang II plus paxilline (1 µM) in VSMC cultures of WKY rats (**A**–**C**) and SHR (**D**–**F**). (**G**) Bar graphs summarizing the VSMC number changes after treatment with the control, Ang II (0.1 µM), and Ang II plus paxilline (1 µM) for 24 hrs. Data are mean ± SE, derived from three experiments and at least triplicate wells in each experiment. * *p* < 0.05 as compared with relative control. ^#^
*p* < 0.05 as compared with WKY Ang II. (**H**). Bar graphs summarizing the cell proliferation measured using WST-1 in VSMCs after treatment as described in the above. Data are mean ± SE, derived from three experiments and at least triplicate wells in each experiment. * *p* < 0.05 as compared with control. ^#^
*p* < 0.05 as compared with WKY Ang II.

**Figure 6 ijms-24-10903-f006:**
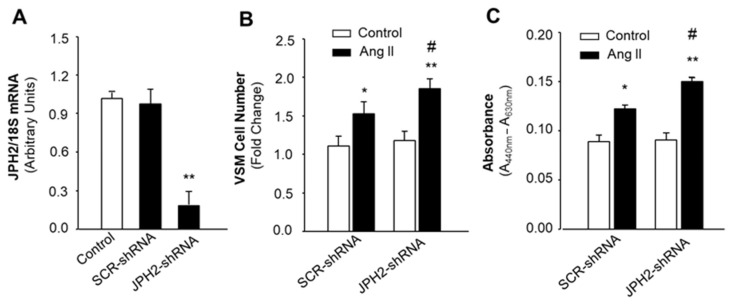
Effect of JPH2 knockdown on Ang II-induced VSMC proliferation. VSMCs were cultured from mesenteric arteries of WKY rats. The cells were treated with saline, AAV2-JPH2-shRNA, or its scramble control (AAV2-SCR-shRNA). (**A**) Bar graphs summarizing JPH2 mRNA levels in VSMCs treated with the control (PBS), AAV2-SCR-shRNA, or AAV2-JPH2-shRNA. Data are mean ± SE, derived from three experiments and triplicate wells in each experiment. ** *p* < 0.01 as compared with PBS control. (**B**) Bar graphs summarizing the change in cell number of VSMCs treated with PBS or Ang II after incubation with AAV2-JPH2-shRNA or AAV2-SCR-shRNA. Data are mean ± SE, derived from three experiments and at least triplicate wells in each experiment. * *p* < 0.05 as compared with relative control. ** *p* < 0.01 as compared with relative control. ^#^
*p* < 0.05 as compared with Ang II plus AAV2-SCR-shRNA. (**C**) Bar graphs summarizing the proliferation of VSMCs treated with PBS or Ang II after incubation with AAV2-JPH2-shRNA or AAV2-SCR-shRNA. Data are mean ± SE, derived from three experiments and at least triplicate wells in each experiment. * *p* < 0.05 as compared with relative control. ** *p* < 0.01 as compared with relative control. ^#^
*p* < 0.05 as compared with Ang II plus AAV2-SCR-shRNA.

**Figure 7 ijms-24-10903-f007:**
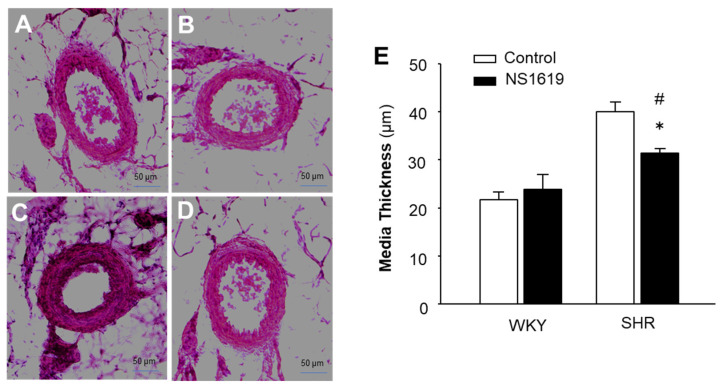
Effect of a BK_Ca_ channel opener on vascular morphology in mesenteric arteries of SHR as compared with WKY rats. The vascular morphology was examined in SHR and WKY rats by H&E staining after treatment with the BK_Ca_ channel opener, NS1619, for 4 weeks (20 µg/kg/day, SQ) or the vehicle control using osmotic minipumps. (**A**–**D**) Micrographs showing representative mesenteric artery cross-sections stained with H&E in WKY rats (**A**,**B**) and SHR (**C**,D) after treatment with the control (**A**,**C**) or NS1619 (**B**,**D**). (**E**) Bar graphs summarizing the media thickness of mesentery arteries of SHR and WKY rats after treatment with the control or NS1619. Data are mean ± SE, derived from 4 sections from each animal and 6 rats from each group. * *p* < 0.05 as compared with relative control. ^#^
*p* < 0.05 as compared with WKY rats treated with NS1619.

## Data Availability

Data are contained within the article.

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
