# Peer review of "Loss of IP3R-BKCa Coupling Is Involved in Vascular Remodeling in Spontaneously Hypertensive Rats"

_ijms, 2023, doi:10.3390/ijms241310903_

Round 1

Reviewer 1 Report

This is a generally well-designed study assessing the effects of IP3R and BK channel coupling in vascular remodeling during hypertension. Although the interaction of IP3R and BK has been well established, the authors provide new insight into how physical coupling between these channels contributes to remodeling in the SHR. Given this unique finding, further characterization of this phenomenon would be beneficial to the manuscript. Additionally, there are a few instances where data presentation can be improved. Specific details are provided below.

Specific Comments:

Major concerns:

1)      The symbols in Figure 3 (panels B and D) differentiating control and AdA are difficult to differentiate. Possible suggestions to improve clarity include increasing symbol size or making one symbol open (instead of filled).

2)      The authors co-IP data suggests a physical uncoupling between IP3R and BK channels, however, a direct interaction is not necessary for SR Ca2+ to enhance BK activity. The additional of immunofluorescence experiments evaluating the relative location of BK and IP3R channels is critical to further the assertion that disruption of the BK/IP3R signaling is critical for reducing BK activity in SHRs.

      Expanding on this concept, and in light of the data presented in figure 6, an evaluation of how the interaction of the sarcolemma and plasma membrane are altered in the SHR should be included. There is a good amount of speculation on this topic in the discussion, but experimental evidence would greatly enhance the interpretation of the results.

3)      Figure 5. In panels G and H there is no significant difference indicated within the Control or SHR groups comparing Ang II + Paxilline vs. Control. Are these groups not statistically different or was this comparison left out for some reason?

4)      The authors should indicate whether there are differences in the sizes of arteries analyzed for vascular remodeling data presented in figure 7. As medial thickness is related to artery size, this information is important to ensure changes in medial thickness are not an artifact of artery size.

5)      As suggested by the authors in the introduction (line 37), hypertension can have various causes.  Do the authors believe that their findings are applicable to all models of hypertensive remodeling or just a subset of causes. Further discussion of this point would be beneficial.

There are a handful of improperly formatted phrases throughout the document, and a few instances of improper verb use.

Author Response

Response to Review 1:

This is a generally well-designed study assessing the effects of IP3R and BK channel coupling in vascular remodeling during hypertension. Although the interaction of IP3R and BK has been well established, the authors provide new insight into how physical coupling between these channels contributes to remodeling in the SHR. Given this unique finding, further characterization of this phenomenon would be beneficial to the manuscript. Additionally, there are a few instances where data presentation can be improved. Specific details are provided below.

Author response: We thank this reviewer for the comments and recommendations, which have improved this manuscript. 

1) The symbols in Figure 3 (panels B and D) differentiating control and AdA are difficult to differentiate. Possible suggestions to improve clarity include increasing symbol size or making one symbol open (instead of filled).

Author response: We appreciate this reviewer’s recommendation. Figure 3 B and D has been updated as the reviewer recommended

2) The authors co-IP data suggests a physical uncoupling between IP3R and BK channels, however, a direct interaction is not necessary for SR Ca2+ to enhance BK activity. The additional of immunofluorescence experiments evaluating the relative location of BK and IP3R channels is critical to further the assertion that disruption of the BK/IP3R signaling is critical for reducing BK activity in SHRs.

      Expanding on this concept, and in light of the data presented in figure 6, an evaluation of how the interaction of the sarcolemma and plasma membrane are altered in the SHR should be included. There is a good amount of speculation on this topic in the discussion, but experimental evidence would greatly enhance the interpretation of the results.

Author response: We absolutely agree with the reviewer’s suggestion that immunofluorescence experiments for coupling between SR IP3R and BK channel on plasma membrane, using Fluorescence Resonance Energy Transfer Technique (FRET) to study the SR Ca2+ -independent mechanism for IP3R to enhance BK activity, is very important data for this research. This is a very good idea and will be the major focus of this research project. In this manuscript, we are reporting that IP3R-BKca channel decoupling is involved in the VSMC proliferation in SHR. However, there are so many unsolved issues in this research project. Currently, we are working on several experiments to answer those questions. For example, what deficits are involved in IP3R-BK decoupling in VSMCs of SHR, Ca2+-dependent or Ca2+-independent mechanism? 2) Why is IP3R-BK coupling disrupted in VSMCs of SHR- is it caused by the dysfunction of Junctophin-2, the tethering protein between sarcolemma and plasma membrane, leading to the dissociation between IP3R and BK channel proteins? We will answer those questions in our future studies. This reviewer’s idea is very helpful for the experimental design and we thank the reviewer for the suggestion.           

3)  Figure 5. In panels G and H there is no significant difference indicated within the Control or SHR groups comparing Ang II + Paxilline vs. Control. Are these groups not statistically different or was this comparison left out for some reason?

Author response: Yes, there are significant differences as compared with control. Figure 5 is now updated accordingly. We thank the reviewer for the recommendation.

4)  The authors should indicate whether there are differences in the sizes of arteries analyzed for vascular remodeling data presented in figure 7. As medial thickness is related to artery size, this information is important to ensure changes in medial thickness are not an artifact of artery size.

Author response: Yes, one sentence has been added to the Methods section (Vascular morphology detection, Page 12, Lines 573-574): “The external diameter of the arteries measured was 100 µm – 150 µm in all arterial cross-sections.” Thanks for this recommendation.   

5)  As suggested by the authors in the introduction (line 37), hypertension can have various causes.  Do the authors believe that their findings are applicable to all models of hypertensive remodeling or just a subset of causes. Further discussion of this point would be beneficial.

Author response: Yes, one paragraph was added to the Discussion section to address this issue (on page 9, lines 418-420) as: “In this study, SHR, a genetic hypertension animal model, was used. Of course, all observations from SHR should be confirmed in other hypertensive animal models, as well as in vascular tissues of hypertensive humans in future studies.” Thank you for this recommendation.

6) There are a handful of improperly formatted phrases throughout the document, and a few instances of improper verb use.

Author response: Yes, the revised manuscript has been edited very carefully. Thanks for this suggestion.  

Reviewer 2 Report

This is an original topic that will be of interest to the readers of the journal. It presents an interesting and carefully designed in vitro investigation of the molecular mechanisms underlying BKCa channel alteration in VSMC phenotype switching and in the development of vascular remodeling during hypertension, by using various techniques. It is generally well written and structured. The manuscript is clear and straight to the point. I have provided few minor revisions below.

Lines 10-11: merit rephrasing

Kindly try to use consistent abbreviation for vascular smooth muscle cells; in some lines it is VSMC, in others it is VSM cells.

Author Response

Review 2

This is an original topic that will be of interest to the readers of the journal. It presents an interesting and carefully designed in vitro investigation of the molecular mechanisms underlying BKCa channel alteration in VSMC phenotype switching and in the development of vascular remodeling during hypertension, by using various techniques. It is generally well written and structured. The manuscript is clear and straight to the point. I have provided few minor revisions below.

Author response: We thank this reviewer for his/her recommendation and suggestions, which helped to improve this manuscript.   

1) Lines 10-11: merit rephrasing

Author response: Yes, this sentence has been updated as: “Vascular smooth muscle cell (VSMC) proliferation and vascular morphology were compared between hypertensive and normotensive rats.” Thanks for this recommendation.

2) Kindly try to use consistent abbreviation for vascular smooth muscle cells; in some lines it is VSMC, in others it is VSM cells.

Author response: Yes, all “VSM cell” have been updated as “VSMC” in the revised manuscript. Thanks for this recommendation.

Round 2

Reviewer 1 Report

This is a generally well-designed study assessing the effects of IP3R and BK channel coupling in vascular remodeling during hypertension. Although the interaction of IP3R and BK has been well established, the authors provide new insight into how physical coupling between these channels contributes to remodeling in the SHR. The authors were highly responsive to reviewer comments.